# Protection of Mild Steel by Waterborne Epoxy Coatings Incorporation of Polypyrrole Nanowires/Graphene Nanocomposites

**DOI:** 10.3390/polym11121998

**Published:** 2019-12-03

**Authors:** Yang Ding, Jiang Zhong, Ping Xie, Jinchuang Rong, Huifang Zhu, Wenbin Zheng, Jinglan Wang, Fei Gao, Liang Shen, Haifeng He, Ziqiang Cheng

**Affiliations:** 1Jiangxi Engineering Laboratory of Waterborne Coating, Department of Coatings and Polymeric Materials, Jiangxi Science and Technology Normal University, Nanchang 330013, China; dingy0128@163.com (Y.D.); ordinaryxp@163.com (P.X.); rjcpolymer@163.com (J.R.); sunnnyzhf@163.com (H.Z.); zhengweb1128@163.com (W.Z.); jxncwjl@163.com (J.W.); feigao2016@jxstnu.com.cn (F.G.); hehf0427@jxstnu.com.cn (H.H.); 2Department of Applied Physics, School of Science, East China Jiaotong University, Nanchang 330013, China; zqcheng_opt@126.com

**Keywords:** polypyrrole nanowires, graphene, waterborne epoxy, corrosion protection

## Abstract

Polypyrrole nanowires/graphene (PPyNG) nanocomposites as anticorrosive fillers were prepared by in situ polymerization in order to improve the anticorrosion performance of waterborne epoxy coatings. Field emission scanning electron microscope (FESEM) and Fourier transform infrared spectroscopy (FTIR) characterized the morphologies and structures of the synthesized PPyNG. The polypyrrole nanowires with about 50 nm in diameter were obtained. Conjugation length of PPy was increased with the addition of graphene. Open circuit potential (OCP) measurements, Tafel polarization curves, and electrochemical impedance spectroscopy (EIS) using an electrochemical workstation evaluated the anticorrosion properties of the waterborne epoxy/PPyNG coatings (EPPyNG). The studied nanocomposite coating possessed superior corrosion protection performance when the graphene content of the filler was 2 wt %. Its corrosion rate was about 100 times lower than that of neat epoxy coating. The higher barrier properties of nanocomposite coating and passivation effect of polypyrrole nanowires were beneficial in corrosion protection.

## 1. Introduction

Metal corrosion is a disturbing phenomenon in which chemical or electrochemical reactions damage metal materials. It would not only result in economic losses, but also threaten the safety of industrial production. Many methods for corrosion protection have been developed, such as environmental modification, anodic protection, cathodic protection, protective coatings, and corrosion inhibitors, or any combination thereof [1,2,3]. Among them, polymer coatings are the most common approach for protecting metal surface from corrosion due to their low cost and high anticorrosion performance. For instance, especially, epoxy coatings have attracted many researchers due to their strong adhesion to substrates and excellent mechanical properties. However, even these coatings fail over prolonged exposure in corrosive media [4]. Many researchers have investigated polymer composites incorporating various functional fillers [5], such as montmorillonite [6], carbon nanotubes [7], graphene [8], and intrinsically conducting polymers (ICPs) [9], to overcome this problem.

ICPs have attracted tremendous attention due to a wide range of potential applications in sensors, supercapacitor electrodes, biological industries, and corrosion protection [10,11]. In the anticorrosion field, the ICPs can be used to form a protective barrier layer and release inhibitors in the coating [12]. Among the available ICPs, polypyrrole (PPy) is the most promising polymer due to its easy polymerization, mechanical stability, and better biocompatibility, as well as tunable electrical property [13]. The electroactive nature of PPy allows for it to oxidize the metal substrate, which results in the formation of a passive oxide layer at the interface between the PPy and underlying metal substrate [14]. PPy is an excellent candidate for replacing hexavalent chromate, since it shows anticorrosion performance that is similar to that found for chromate [6,15]. Loading PPy can also enhance the corrosion inhibiting properties of zinc-filled epoxy coatings [16,17]. The surface treatment of pigment with PPy is found to be beneficial to the anticorrosion and mechanical properties of the epoxy paint [18].

Despite these advantages, the aggregation tendency of PPy particles is a major restriction of the processing and application of epoxy/PPy coating. Integration with montmorillonite and graphene turns out to be an effective method for avoiding aggregations and improve dispersion of PPy in the polymer matrix, which contributes to a great corrosion protection enhancement [6,19,20]. Graphene has higher aspect ratio than clay platelet, which can simultaneously improve not only the barrier properties, but also several mechanical, functional, and thermal properties of epoxy coatings [21,22]. The synergistic effect of PPy/graphene greatly enhances the anticorrosion performance of epoxy coating [23].

PPy with one-dimensional nanostructure, such as nanotube [24], nanorod [25], and nanowire [26], are used in various applications including sensors, supercapacitors and battery electrodes [13,27,28]. Up to now, there has been no report on using PPy nanowire as additive for waterborne anticorrosion coatings. In this paper, the PPy nanowires (PPyN) were prepared by a soft template method. The PPyN/graphene (PPyNG) nanocomposites were synthesized through in situ polymerization. The molecule structures and morphologies of the PPyNG nanocomposites were also analyzed. After that, the PPyNG nanocomposites were incorporated into waterborne epoxy coatings to study their anticorrosion performance for mild steel. For the convenience discussion, the present nanocomposite referred to as PPyNG, and nanocomposite coating corresponded to the epoxy/PPyNG coating.

## 2. Materials and Methods

### 2.1. Materials

The pyrrole monomer was purchased from Aladdin Bio-Chem Technology Co., Ltd. (Shanghai, China). Ammonium persulfate (APS), potassium permanganate (KMnO_4_), sulfuric acid (H_2_SO_4_), hydrogen peroxide (H_2_O_2_), N-methyl pyrrolidone (NMP), Sodium chloride (NaCl), and cetyltrimethylammonium bromide (CTAB) were obtained from Sinopharm Chemical Reagent Co., Ltd. (Shanghai, China). The flake graphite powder (325 mesh) was purchased from Sigma-Aldrich Chemicals (Shanghai, China). All of the chemicals were of analytical reagent grade and used without further purification. Hexion Inc. (Columbus, Ohio, US) provided waterborne epoxy resin (Epikote 6520-WH-53) and curing agent (Epikur 8538-Y-68). The Q235 mild steels (C: 0.14%, Mn: 0.3%, S: 0.05%, P: 0.045%, and rest being Fe) with area of 42 mm × 10 mm were purchased from Biuged Laboratory Instruments (Guangzhou) Co., Ltd. (Guangzhou, China). The mild steels were polished while using 400 grift sand papers and then cleaned in ethanol and acetone.

### 2.2. Preparation of Graphene

Graphene was prepared by following the method by Dong et al. [29]. In brief, 100 g KMnO_4_ (1 wt % equiv.) was added in batches into concentrated H_2_SO_4_ (2 L, 98%) over a period of 45 min. in an ice-water bath. 100 g graphite (1 wt % equiv., 325 mesh) was then added in batches under stirring at 35 °C for 2 h. The black flakes were filtered through a 200-mesh sieve and poured into 2 L of ice water after reaction. Subsequently, 30 wt % H_2_O_2_ was added to decompose the insoluble manganese dioxide. The wet powders of pretreated graphite were obtained after filtering and washing. Pretreated graphite was then dispersed in the alkali water (pH = 14) under ultrasonication while using a sonic vibra-cell VC505 processor in 60% power for 1 h, resulting in black graphene slurry. The graphene slurry was centrifuged at 10,000 rpm for 10 min. and then repeatedly washed by a large amount of water (4–6 times) until pH approached 10. The prepared graphene was re-dispersed in N-methyl pyrrolidone (NMP) for further use.

### 2.3. Synthesis of PPy Nanowires/Graphene Nanocomposites

The PPy nanowires/graphene nanocomposites (PPyNG) were synthesized through in situ chemical oxidative polymerization. In a typical process, 0.31 mL pyrrole monomer and 0.91 g CTAB were added in a mixture of 125 mL 0.2 M hydrochloric acid and graphene-NMP dispersion and stirred for 2 h at ambient temperature, and then cooled to 0–5 °C. A precooled aqueous solution of 10 mL 0.2 M HCl containing 4 mmol APS was added into above solution in batches. The reaction was allowed to proceed under stirring for 4 h at about 0–5 °C. The resulting product was filtered and rinsed with deionized water and ethanol (4–6 times) until the filtrate was colorless. Finally, the product was dried in a vacuum oven at 40 °C for 24 h. The weight ratio of pyrrole to graphene varied as 99:1, 98:2, 97:3, and the resulting black composites were named as PPyNG1, PPyNG2, and PPyNG3, respectively. For comparison, neat PPy nanowires (PPyN) were fabricated by similar method absence of graphene dispersion.

### 2.4. Fabrication of Nanocomposites Coatings

The investigated coatings in this study consisted of the epoxy without the addition of filler as well as with the incorporation of PPy nanowires and PPy nanowires/graphene as the functional additives. Three weight concentrations of PPy nanowires additives had been prepared in order to achieve the optimal level of additives, and their anticorrosion properties are shown in Appendix A, Appendix A and Appendix A. The 0.5% additive based on the total formulation was finally determined. The preparation procedure was as follows. Firstly, the PPy nanowires and PPyNG nanocomposites were completely dispersed in 10 mL of deionized water with a high-speed dispersion and then added into 30 g waterborne epoxy resin. Next, 10 g curing agent was added into the above dispersive media and the mixture was painted on the pretreated mild steels. The resulting coating was obtained after curing for 48 h at room temperature. The prepared epoxy coatings that were loaded with PPy nanowires, PPyNG1, PPyNG2, and PPyNG3 were denoted as EPPyN, EPPyNG1, EPPyNG2 and EPPyNG3, respectively. The dry coating thickness was around 37 μm. For comparison, the neat epoxy coating was also prepared through a similar method without loadings, which was named Blank. Figure 1 shows the preparation process for the fabrication of waterborne epoxy/PPyNG coatings (EPPyNG) coating materials.

### 2.5. Characterization

The Fourier transform infrared spectrometer (FTIR) spectra of PPyNG films were collected by a Bruker-Veretex70 spectrometer (Bruker Company, Karlsruhe, Germany) while using KBr pellets. The scanning electron microscopy (SEM) images of PPyN and PPyNG composites were obtained while using a Zeiss Sigma FE-SEM (Carl Zeiss AG, Geraniah, Germany). The electrochemical measurements were carried out to characterize the anticorrosive properties of blank epoxy, EPPyN and EPPyNG coatings using a CHI 660E electrochemical workstation (Chinstruments Co., Ltd., Shanghai, China) that was equipped with a conventional three-electrode cell with a saturated calomel electrode (SCE) as reference, a platinum counter electrode with 1 cm^2^ area, and a working electrode. The bare mild steels or mild steels with coatings were sealed while using sealant (paraffin: Rosin = 1:1) and Teflon to leave 1 cm^2^ area opening to the electrolytic solution. The electrolyte was 3.5 wt % NaCl solution. Open-circuit potential (OCP) was recorded for the blank epoxy, EPPyN and EPPyNG coatings up to 30 days of immersion time. The potential dynamic polarization curves of blank epoxy, EPPyN, and EPPyNG coatings were performed with a sweep rate of 2 mV/s from the cathodic direction to the anodic direction. The electrochemical impedance spectroscopy (EIS) measurements were collected in the frequency range of 100 kHz to 0.01 Hz while using an alternating current signal with the amplitude of 5 mV. All of the electrochemical tests were conducted at room temperature. A ZQ-401 microscope was used to record the optical microscopic images of EPPyN coatings (Zhiqi Co., Ltd., Shanghai, China).

## 3. Results

Figure 2 shows the FTIR spectra of the PPyN and PPyNG nanocomposites. The broad band at 3000–3500 cm^−1^ arose from N–H stretching vibrations [30]. The characteristic polypyrrole peaks located at 1558 and 1478 cm^−1^ were due to the asymmetric and symmetric ring-stretching modes, respectively [31]. The bands at 1048 and 1321 cm^−1^ were attributed to C–H deformation vibrations and C–N stretching vibrations, respectively [32]. In addition, the peaks that were centered at 1202 and 923 cm^−1^ were assigned to the doping states of PPy [33]. When comparing to the FTIR spectra of PPyN, all peaks had also appeared in the PPyNG nanocomposites. It was worth noting that the C–N stretching vibrations peak of PPyNG nanocomposites had been downshifted to 1310 cm^−1^, which was probably due to the π–π interactions between graphene layers and aromatic polypyrrole rings [34]. Moreover, the ratio between the peak area of the skeletal band 1478 cm^−1^ and oxidization state sensitive 1558 cm^−1^ band (*I*_1478_/*I*_1558_) could be used to calculate the conjugation length [35,36]. The ratio was 0.1304, 0.1328, 0.1427, and 0.1515 for PPy, PPyNG1, PPyNG2, and PPyNG3, respectively, which suggested that the conjugation length increased with further incorporation of the graphene. It might be ascribed to the fact that the strong interfacial interaction between PPy and graphene induced more electrons that were delocalized either in the pyrrole units or in the benzene ring units of graphene [35,37].

Figure 3 shows the morphologies of pure PPyN and the PPyNG nanocomposites. The SEM image (Figure 3a) revealed the uniform nanowire structure with an average diameter of about 50 nm and length of several micrometers. For PPyNG1 and PPyNG2 nanocomposites (Figure 3b,c), PPy held its wire-like morphology with a similar size to pure PPyN. However, from Figure 3d, the granular morphology of PPyNG3 was observed. The isolated graphene nanosheets could also be seen due to increasing concentration of graphene nanosheets. The result of the SEM images indicated that the morphology of PPy nanowires was affected by the higher content graphene. Similar phenomenon was observed in other’s research [38].

Figure 4 shows the evolution of open circuit potential (OCP) for coatings on mild steels in a 3.5% NaCl corrosive solution. The monitoring of OCP allowed for the assessment of the inclination of corrosion [39]. In the initial immersion, the OCP values exhibited a less negative potential for the Blank, EPPyN, and EPPyNG3 coatings, (−0.3 to −0.45 V), but the EPPyNG2 showed a more noble OCP value of around −0.14 V. This relative high potential demonstrated that EPPyNG2 provided excellent protective performance. The OCP value tended to decrease with the immersion time for all of the studied coatings. However, the lower rate of declination was observed in EPPyN, EPPyNG1, EPPyNG2, and EPPyNG3 when compared to the blank epoxy coating after immersion 20 days. It indicated that the oxidative PPy functional fillers passivated the steel, resulting in higher OCP [18]. The passivation layer effectively prevented the steel from corrosion. A few slight increases were observed for Blank and EPPyNG1 after immersion several days, probably due to the accumulation of corrosion products [23]. The potential of EPPyNG3 coating dropped sharply to −0.70 V after immersion 20 days, revealing that excess graphene was detrimental to the anticorrosion property [40].

The potentiodynamic polarization tests were carried out to study the anticorrosion performance of EPPyNG composites coatings. Figure 5 shows the Tafel curves of the as prepared coatings immersion in corrosive solutions. Table 1 shows the electrochemical parameters that were obtained from Tafel curves. The corrosion current density (*I*_corr_), corrosion potential (*E*_corr_), anodic Tafel slope (*b*_a_), and cathodic Tafel slope (*b*_c_) were estimated from the Tafel extrapolation of anodic and cathodic lines to the point of intersection. The more negative *E*_corr_ and larger *I*_corr_ suggested a faster corrosion rate, while the more positive *E*_corr_ and the smaller *I*_corr_ indicated a slower corrosion process [41]. The *E*_corr_ of the neat epoxy resin coated mild steel was −784 mV. A significantly positive shift *E*_corr_ of −565 mV was obtained for the composites coating with PPyN, confirming that EPPyN engaged in redox reactions, resulting in the formation of metal oxide passive layer [20,42]. By addition of PPyNG, the *E*_corr_ dramatically increased to −537 mV with 1 wt % graphene and −482 mV with 2 wt % graphene. Advanced corrosion protection effect of EPPyNG coating compared to EPPyN might arose from dispersing graphene nanosheets to increase the tortuosity of the diffusion pathway of H_2_O, O_2_, and Cl^−^ [43]. However, the coating with excess graphene (EPPyNG3) exhibited lower *E*_corr_ than EPPyNG2, probably because larger fraction of graphene affected the growth of PPy nanowires and induced the defects in PPyNG3 composites [38], which was in agreement with the results of SEM. Furthermore, the higher *b*_a_/*b*_c_ ratio of EPPyNG2 was observed, which revealed a reduction of the anodic dissolution [23]. The *I*_corr_ values of EPPyNG1 and EPPyNG2 coated mild steel considerably decreased when compared with EPPyN. For the EPPyNG2 coatings, the *I*_corr_ was lowest than other coatings, which indicated the best anticorrosion performance in all of the studied samples, which corresponded to results of OCP tests.

The quantitative analysis of Tafel curve was also investigated, and Table 1 summarizes the result. The corrosion rate, *v*_corr_ (mm/year) was obtained from Equation (1) [44]:(1)$$v_{\mathrm{corr}}=\frac{MI_{\mathrm{corr}}}{n\rho }\times 3270$$
where the molecular weight (*M*) is 55.85 g/mol for Q235, *I*_corr_ is the corrosion current density (A/cm^2^), *n* is 2 for the oxidation of steel, and the density (*ρ*) is 7.85 g/cm^3^ for Q235, 3270 is a constant. *R*_p_ is the polarization resistance calculated by the slope of the polarization curve at the *E*_corr_ according to the Stern–Geary Equation (2) [45]:(2)$$R_{p}=\frac{b_{a}b_{c}}{2.303\left(b_{a}+b_{c}\right)I_{\mathrm{corr}}}$$

Here, *b*_a_ and *b*_c_ are the anodic and cathodic Tafel slopes, *I*_corr_ is the corrosion current density. The protection efficiency (PE, %) was calculated via Equation (3) [46]:(3)$$\mathrm{PE}=\frac{R_{p}^{-1}\left(bare\right)-R_{p}^{-1}\left(coated\right)}{R_{p}^{-1}\left(bare\right)}\times 100\%$$
where *R_p_*(*bare*) and *R_p_*(*coated*) denote the polarization resistance of bare and coated steel, respectively. Appendix A shows the Tafel curve of bare steel.

The calculated *v*_corr_ value of 8.9 × 10^−5^ mm/year for EPPyNG2 was about 100 times lower than that for blank epoxy coating. The result of *v*_corr_ demonstrated that the EPPyNG coatings displayed higher anticorrosion performance than epoxy coating with PPyN alone, and the best performance was achieved when the graphene-doped ratio was 2 wt %. The mild steel that was coated with EPPyNG2 exhibited a *R*_p_ value of 5.6 × 10^6^ Ωcm^2^, which was higher than that of bare mild steel, neat epoxy, EPPyN, EPPyNG1, and EPPyNG3 coated ones. In the case of protection efficiency, the highest PE value from EPPyNG2 (99.9%) described that PPyNG2 loading in the epoxy coating can provide superior inhibition corrosion performance.

EIS is a powerful tool for investigating the corrosion protection of the coatings [47]. Figure 6 shows the Nyquist and Bode plots of the coated mild steels during different immersion days. For neat epoxy coating (Figure 6a), the Nyquist plots displayed two capacitive arcs after 15 days immersion, which indicated that the electrolyte was in contact with the metal surface. The first semicircle at high frequency region and the second part at middle-low frequency region were due to the impedance of coating and corrosion reactions, respectively [48]. The radius of capacitive impedance loop in high-frequency domain diminished during the immersion, which implied declined corrosion protective properties for mild steel. The EPPyN (Figure 6c), EPPyNG1 (Figure 6g), and EPPyNG3 (Figure 6i) showed the same trend. However, for the EPPyNG2 coating, the radius of capacitive impedance arc at low frequency region in the immersion seven days suddenly expanded beyond that of three days, which suggested that PPy might react to passivate the metal substrate [23]. The second semicircle was also observed after long time immersion for EPPyNG2, but its radius was significantly smaller than that of other coatings, which indicated less response from the pitting corrosion of the metal substrate [43].

The impedance modulus at low frequency (|Z|_0.01Hz_) was used as a semi-quantitative standard of coating’s protective performance for the Bode patterns [49], as summarized in Table 2. The impedance modulus dramatically decreased after 15 days and then progressively reduced with the increasing immersion time for all studied coatings, probably owing to the penetration of water and movement of ions through the coatings [50]. The |Z|_0.01Hz_ of EPPyNG2 (Figure 6h), in the early immersion time, was up to 7.7 × 10^6^ Ωcm^2^, higher than those of neat epoxy (1.0 × 10^6^ Ωcm^2^). The EPPyNG2 coating had better barrier properties when compared to neat epoxy, because the well dispersed PPyNG could fill the structural and pinhole porosity of neat epoxy and then inhibited the water penetration. The conjugation length of EPPyNG2 was longer than those of EPPyN and EPPyNG1, which resulted in electrons easier delocalization and in favor of the formation of passivation layers [51,52]. Hence, the EPPyNG2 also exhibited better anticorrosion performance than those of EPPyN (4.5 × 10^6^ Ωcm^2^) and EPPyNG1 (5.4 × 10^6^ Ωcm^2^). Although EPPyNG3 possessed longer conjugation length, its excess graphene content might increase the defects due to influence of polypyrrole nanowires, and probably generated micro galvanic corrosion that showed the corrosion-promotion activity [53]. Therefore, the EPPyNG3 (1.6 × 10^6^ Ωcm^2^) displayed worse anticorrosive property when compared with EPPyNG2. In addition, the impedance modulus of EPPyNG2 remained 1.2 × 10^5^ Ωcm^2^ after 30 days, which was higher than other coatings.

Moreover, ZSimpWin software further fitted the EIS measurements while using the equivalent electric circuits, as shown in Figure 7. The electrical equivalent circuits were composed of *R*_s_, *R*_pore_, *R*_ct_, *Z*_w_, *Q*_c_, and *Q*_dl_, which represented the solution resistance, pore resistance, charge-transfer resistance, Warburg impedance, and the constant phase elements substituted for the coating capacitance (*C*_c_) and double-layer (steel/solution) capacitance (*C*_dl_), respectively. *R*_pore_ modeled ionically conducting paths across the coating, which could be used to evaluate the barrier performance of the coatings [54]. Models (a) and (b) were fitted with the EIS data of pure epoxy coating and composites coatings, respectively.

Figure 8 shows the time-dependent behavior of *R*_pore_ and *C*_c_ on the basis of the above fitting models. Generally, a higher *R*_pore_ and lower *C*_c_ suggested that a small amount of corrosive media penetrated into the coatings [55]. It could be observed that the *R*_pore_ of neat epoxy coating gradually decreased from 7.3 × 10^5^ Ωcm^2^ to a much lower value (1.3 × 10^4^ Ωcm^2^). The *R*_pore_ of coatings with EPPyN, EPPyNG1, and EPPyNG3 decreased to some extent, while the EPPyNG2 coating always maintained much higher values. The *C*_c_ of pure epoxy coating continuously increased from 3.1 × 10^−9^ F/cm^2^ to 1.1 × 10^−8^ F/cm^2^. In contrast, the EPPyNG2 displayed a much lower *C*_c_ value when compared to those of neat epoxy and EPPyN coating. The EPPyNG3 coating also exhibited lower *R*_pore_ and higher *C*_c_ than EPPyNG2, which confirmed that excess graphene induced the defect coating and reduced corrosion protection. Consequently, it could be concluded that the EPPyNG2 composite coating showed the outstanding corrosion resistance, which was in agreement with the potentiodynamic polarization results.

According to the above results, Figure 9 demonstrates the mechanism of corrosion protection for the mild steel substrate with neat epoxy coating and PPyNG nanocomposite coatings. For pure epoxy coating, corrosive mediums (H_2_O, O_2_, and Cl^-^, etc.) could penetrate the coating easily due to the minute crevices of the surface. However, the well-dispersed PPyNG composite, as an anti-corrosion barrier, repaired the cracks of epoxy coating, and enhanced the tortuosity of the diffusion pathway at a great extent. Furthermore, PPyN reacted with steel to form a dense layer of passive oxide film, which resulted in substantially reduced penetration of corrosive medias. Hence, the anti-corrosion performance of the EPPyNG nanocomposite coatings was significantly improved.

## 4. Conclusions

In this study, the PPy nanowires were synthesized while using CTAB as soft template, and combined with graphene nanosheets by in situ oxidation polymerization. The incorporation of PPy nanowires/graphene into waterborne epoxy on mild steel substrate was performed. Potentiodynamic polarization plots, impedance measurements, and fitted *R*_pore_ and *C*_c_ were used to study the anticorrosion performance of the coatings. The PPy nanowires/graphene nanocomposite coating exhibited improved anticorrosion performance in comparison with pure epoxy and epoxy/PPy nanowires samples. The passivation effect of the PPy nanowires contributed to the effective inhibiting corrosive effect. Meanwhile, the well-dispersed PPyNG could block the coating pores and decrease the corrosive medias diffusion toward the substrate. Therefore, PPy nanowires/graphene nanocomposite would be used as a promising anticorrosion pigment.

## Figures and Tables

**Figure 1 polymers-11-01998-f001:**
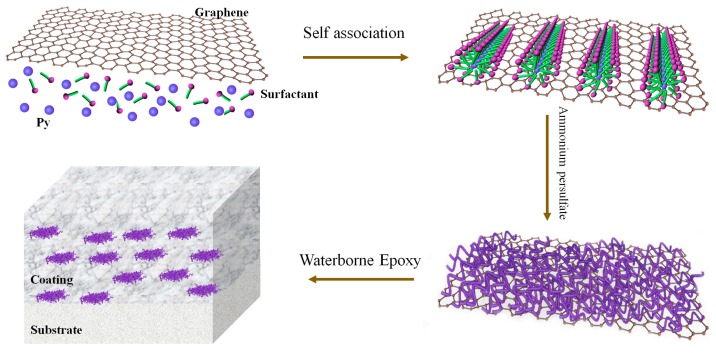
Schematic illustration of the preparation of waterborne epoxy/PPyNG coatings (EPPyNG) nanocomposite coatings.

**Figure 2 polymers-11-01998-f002:**
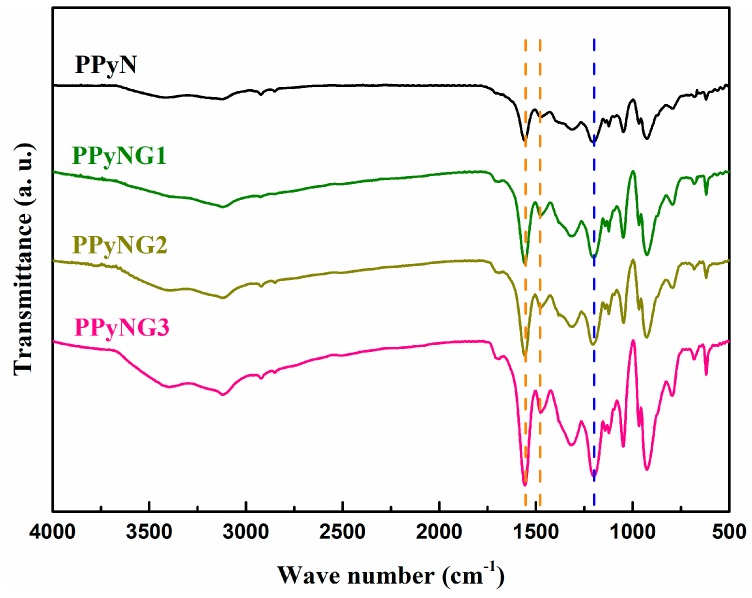
Fourier transform infrared spectroscopy (FTIR) spectrum of the PPyN, PPyNG1, PPyNG2, and PPyNG3.

**Figure 3 polymers-11-01998-f003:**
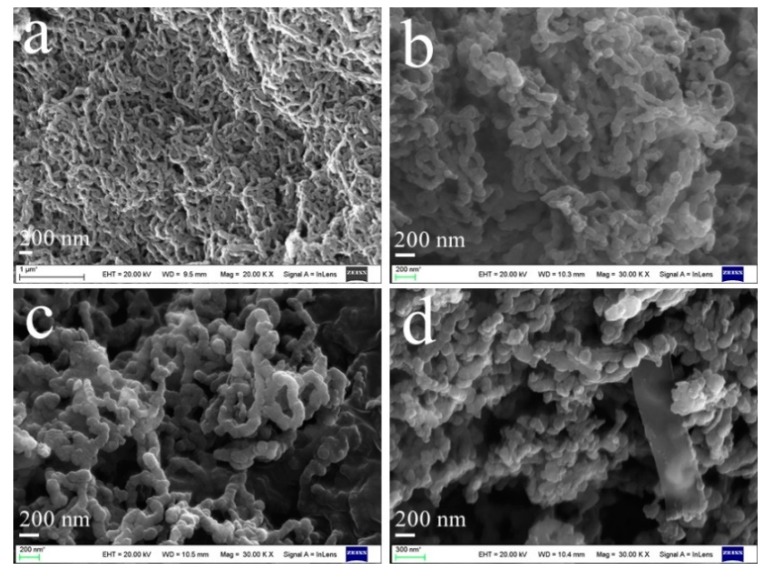
Scanning electron microscopy (SEM) images of the (**a**) PPyN, (**b**)PPyNG1, (**c**)PPyNG2, and (**d**) PPyNG3.

**Figure 4 polymers-11-01998-f004:**
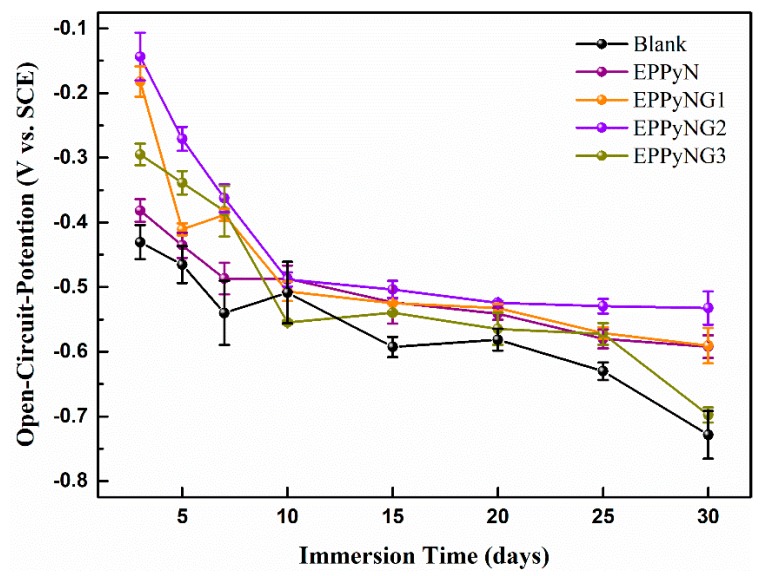
Evolution of open circuit potential (OCP) value for the samples with continuous immersion.

**Figure 5 polymers-11-01998-f005:**
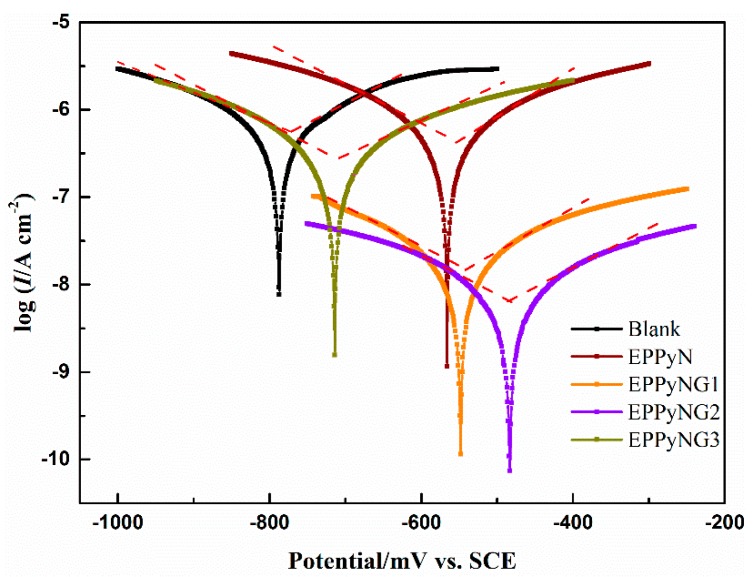
Tafel curves of coated mild steel.

**Figure 6 polymers-11-01998-f006:**
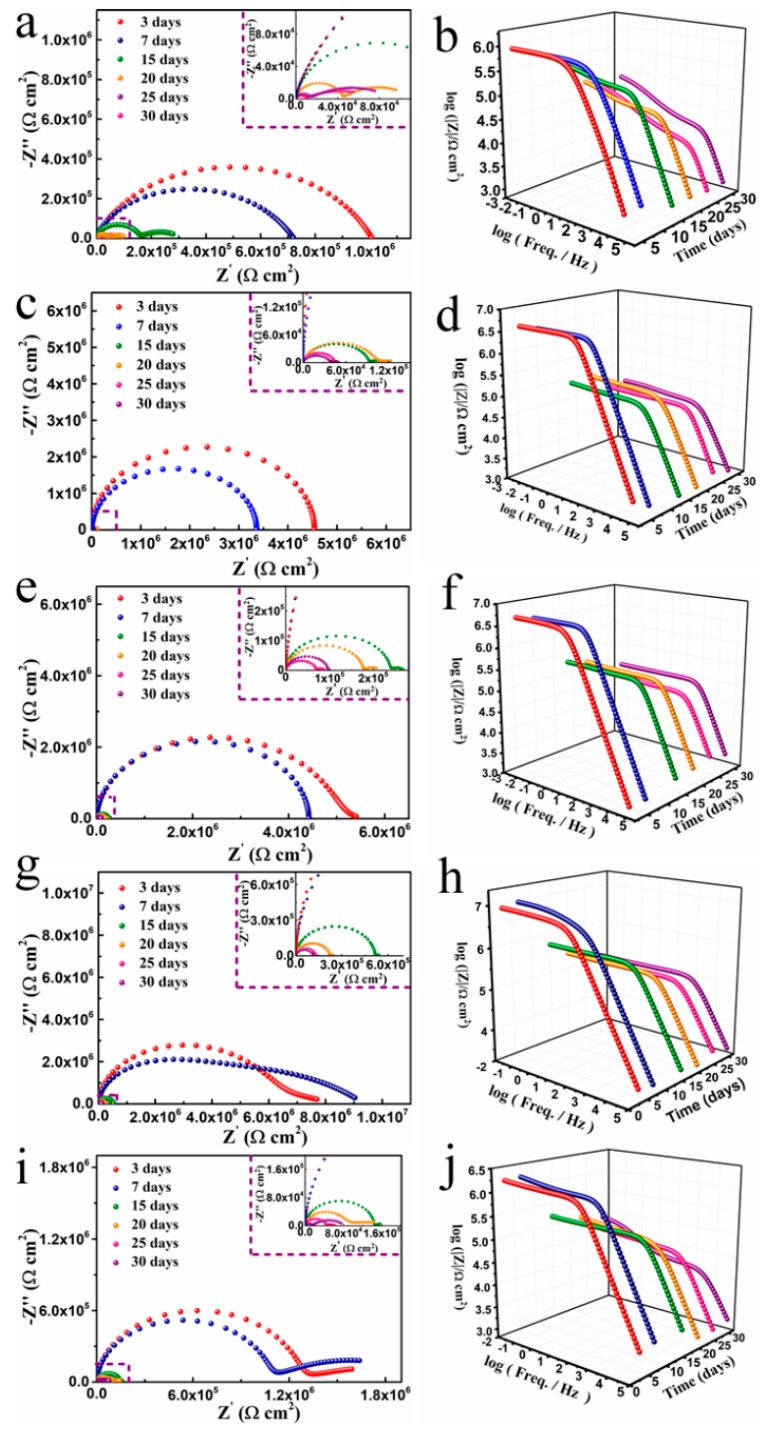
Nyquist and Bode plots of the coatings: (**a**,**b**) blank, (**c**,**d**) EPPyN, (**e**,**f**) EPPyNG1, (**g**,**h**) EPPyN2, and (**i**,**j**) EPPyNG3.

**Figure 7 polymers-11-01998-f007:**
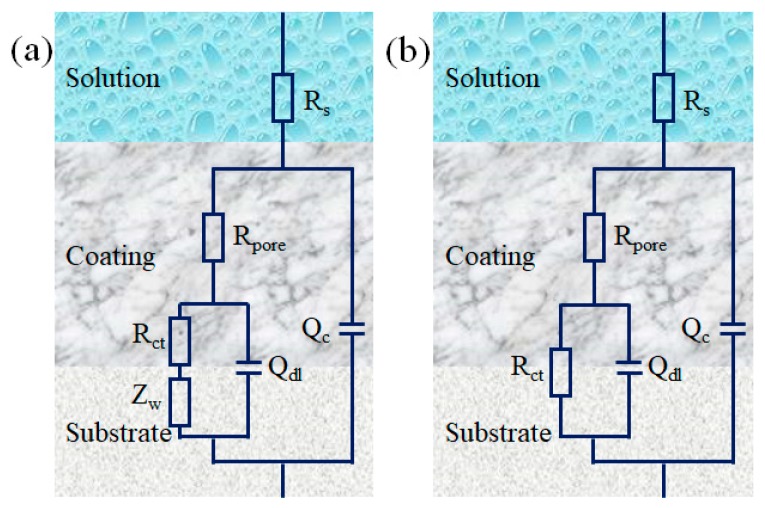
(**a**,**b**) Equivalent electrical circuits of the coatings.

**Figure 8 polymers-11-01998-f008:**
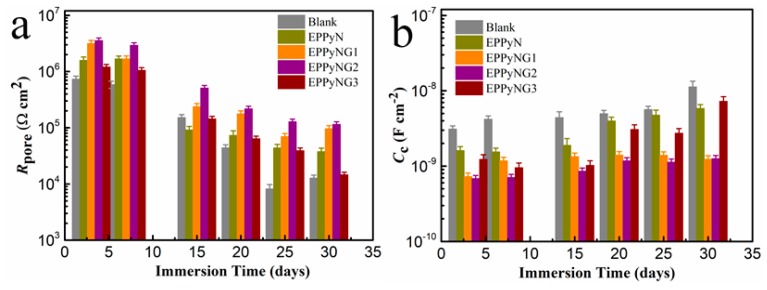
(**a**) *R*_pore_ and (**b**) *C*_c_ values with immersion in 3.5 wt % NaCl solution.

**Figure 9 polymers-11-01998-f009:**
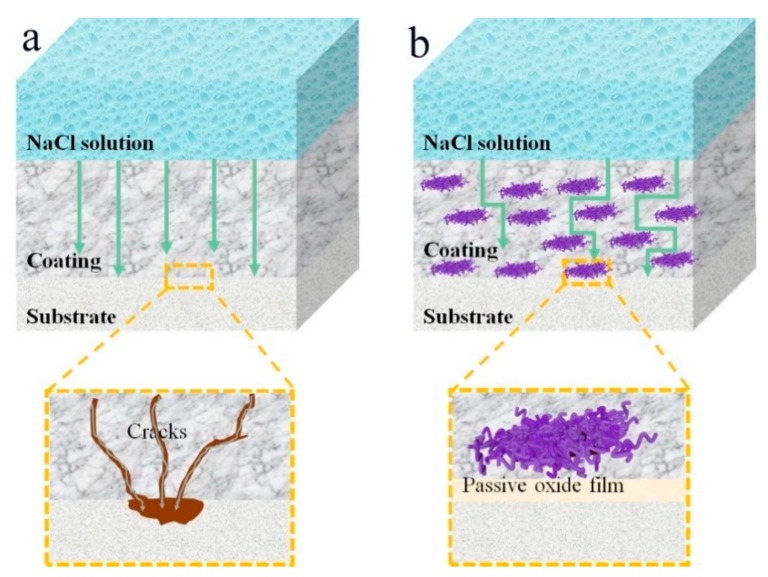
Schematic illustration of corrosion protection of the mild steel with (**a**) neat epoxy coating and (**b**) EPPyNG composite coating.

**Table 1 polymers-11-01998-t001:** Parameters of Tafel polarization curve for coated Q235 substrates.

Sample	*E*_corr_ (mV)	*I*_corr_ (A/cm^2^)	*b*_a_ (mV/dec)	*b_c_* (mV/dec)	*v*_corr_ (mm/year)	R_p_ (Ω cm^2^)	PE (%)
**Bare**	−1019	1.1 × 10^−5^	170.9	−173.5	0.13	3.4 × 10^3^	-
**Blank**	−784	7.6 × 10^−7^	193.8	−221.8	8.8 × 10^−3^	5.9 × 10^4^	94.3
**EPPyN**	−565	5.7 × 10^−7^	201.4	−195.3	6.6 × 10^−3^	7.6 × 10^4^	95.5
**EPPyNG1**	−537	1.9 × 10^−8^	209.1	−192.1	2.2 × 10^−4^	2.3 × 10^6^	99.8
**EPPyNG2**	−482	7.7 × 10^−9^	203.9	−193.9	8.9 × 10^−5^	5.6 × 10^6^	99.9
**EPPyNG3**	−713	3.0 × 10^−7^	211.8	−186.1	3.5 × 10^−3^	1.4 × 10^5^	97.6

**Table 2 polymers-11-01998-t002:** Results of electrochemical impedance spectroscopy (EIS) for coated Q235 substrates.

Sample	|Z|_0.01Hz_ (Ωcm^2^)					
	3 days	7 days	15 days	20 days	25 days	30 days
**Blank**	1.0 × 10^6^	7.2 × 10^5^	2.8 × 10^5^	9.6 × 10^4^	5.5 × 10^4^	7.6 × 10^4^
**EPPyN**	4.5 × 10^6^	3.4 × 10^6^	1.1 × 10^5^	1.2 × 10^5^	4.7 × 10^4^	5.0 × 10^4^
**EPPyNG1**	5.4 × 10^6^	4.4 × 10^6^	2.7 × 10^5^	2.1 × 10^5^	8.9 × 10^4^	1.0 × 10^5^
**EPPyNG2**	7.7 × 10^6^	9.0 × 10^6^	5.4 × 10^5^	2.4 × 10^5^	1.4 × 10^5^	1.2 × 10^5^
**EPPyNG3**	1.6 × 10^6^	1.6 × 10^6^	1.6 × 10^5^	1.4 × 10^5^	6.2 × 10^4^	7.6 × 10^4^

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
