# Peer review of "Protection of Mild Steel by Waterborne Epoxy Coatings Incorporation of Polypyrrole Nanowires/Graphene Nanocomposites"

_polymers, 2019, doi:10.3390/polym11121998_

Round 1
Reviewer 1 Report
Page 1 line 28-35: “The introduction should briefly place the study in a broad context and highlight why it is important. It should define the purpose of the work and its significance. The current state of the 29 research field should be reviewed carefully and key publications cited. Please highlight controversial 30 and diverging hypotheses when necessary. Finally, briefly mention the main aim of the work and 31 highlight the principal conclusions. As far as possible, please keep the introduction comprehensible 32 to scientists outside your particular field of research. References should be numbered in order of 33 appearance and indicated by a numeral or numerals in square brackets, e.g., [1] or [2,3], or [4–6]. See 34 the end of the document for further details on references.” THIS IS THE DESCRIPTION HOW AN INTRODUCTION SHOULD BE COMPOSED, PRESCRIBED BY THE JOURNAL!!!!
Page 10 line 291: “Authors should discuss the results and how they can be interpreted in perspective of previous studies and of the working hypotheses. The findings and their implications should be discussed in the broadest context possible. Future research directions may also be highlighted.” THESE SENTENCES DO NOT BELONG TO THE SUBMISSION BUT WAS GIVEN AS ADVICE FOR THE AUTHORS!!!!
Please, use the correct full name of the abbreviations: page 1 line 17 “Fourier transform spectroscopy (FTIR)” (Fourier transform infrared spectroscopy); page 1 line 20: “ electrochemical impedance spectra (EIS) (electrochemical impedance spectroscopy as it is indicated correctly on page 4 line 138).
Page 1 line 35: “abominable” This is not the best attribute, please, replace it.
Page 2 line 93: “was then dispersed in added to the alkali water” Please, rephrase this part of the sentence; either “dispersed in..” or “…added to…”
Page 3 line 95: “was obtained through centrifuged at 10,000 rpm” ” Please, rephrase this part of the sentence (e.g.: …was centrifuged at..).
Page 3 line 113: “The coatings investigated in this study consisted in the epoxy based without the addition of filler….” Please, rephrase this part of the sentence.
Page 3 line 121: “…and the mixture was painted on the pretreated mild steel” Please, inform the readers about the pretreatment of the mild steel.
It is interesting that the Authors used NaCl solution in the electrochemical experiments as this chemical causes pitting corrosion and not general one.
How can the Authors prove that during the PPy-graphen preparation the graphene was not oxidized? They mention: “The PPy nanowires/graphene nanocomposites (PPyNG) were synthesized through in situ chemical oxidative polymerization.”
In Figure 3 it would be better to use the same scale (instead of 1μm, 200 nm, 300 nm).
Page 7 line 246: “The EPPyNG2 coating had better barrier property compared to neat epoxy and EPPyN, because the well dispersed PPyNG could fill the structural and pinhole porosity of neat epoxy and then inhibited the water penetration.” Though the Authors described and proved the fact that the 2% of PPyNG2 composite coating showed higher anticorrosion activity than the coating with 3%, but they did not give satisfying explanation for this phenomenon. It is not supported by the structure of the PPyNG1 and PPyNG2, which seemed to be very similar shown in the Figure 3 but the PPyNG3 had a different structure that shows the differences in the efficiency is not due to the structure. On page 9 line 274 the Authors say: “The EPPyNG3 coating also exhibited lower Rpore and higher Cc than EPPyNG2, confirming that excess graphene induced the defect coating and reduced corrosion protection.“ This statement specially needs explanation concerning the sentence given on page 9 line 284: “PPyN reacted with steel to form a dense layer of passive oxide film, resulting in substantially reduced penetration of corrosive medias.” One can think that in the presence of more PPyNG the formation of the passive film is more likely.
Author Response
Dear Reviewer 1:
Special thanks to you for your good comment. Your comments are all valuable and very helpful for revising and improving our paper, as well as the important guiding significance to our researchers. We have studied comments carefully and have made correction. The amendments are highlighted in yellow in the revised manuscript. We would like to resubmit it for your kind consideration. Point by point responses to your comments are listed below this letter.
We hope that the revised version of the manuscript is now acceptable for publication. Looking forward to hearing from you.
Thank you and best regards.
Yours sincerely,
Jiang Zhong, Ph. D.
jiangzhong@jxstnu.com.cn
Page 1 line 28-35: “The introduction should briefly place the study in a broad context and highlight why it is important. It should define the purpose of the work and its significance. The current state of the 29 research field should be reviewed carefully and key publications cited. Please highlight controversial 30 and diverging hypotheses when necessary. Finally, briefly mention the main aim of the work and 31 highlight the principal conclusions. As far as possible, please keep the introduction comprehensible 32 to scientists outside your particular field of research. References should be numbered in order of 33 appearance and indicated by a numeral or numerals in square brackets, e.g., [1] or [2,3], or [4–6]. See 34 the end of the document for further details on references.” THIS IS THE DESCRIPTION HOW AN INTRODUCTION SHOULD BE COMPOSED, PRESCRIBED BY THE JOURNAL!!!!
Response: We are very sorry for our sloppy preparation of manuscripts with the template of the journal. We have deleted these sentences in the Introduction.
Page 10 line 291: “Authors should discuss the results and how they can be interpreted in perspective of previous studies and of the working hypotheses. The findings and their implications should be discussed in the broadest context possible. Future research directions may also be highlighted.” THESE SENTENCES DO NOT BELONG TO THE SUBMISSION BUT WAS GIVEN AS ADVICE FOR THE AUTHORS!!!!
Response: We apologize for our carelessness. These sentences are deleted in the Conclusion section.
Please, use the correct full name of the abbreviations: page 1 line 17 “Fourier transform spectroscopy (FTIR)” (Fourier transform infrared spectroscopy); page 1 line 20: “ electrochemical impedance spectra (EIS) (electrochemical impedance spectroscopy as it is indicated correctly on page 4 line 138).
Response: The full name of FTIR and EIS have been corrected.
Page 1 line 35: “abominable” This is not the best attribute, please, replace it.
Response: We have replaced “abominable” with “disturbing”.
Page 2 line 93: “was then dispersed in added to the alkali water” Please, rephrase this part of the sentence; either “dispersed in..” or “…added to…”
Response: This redundant phrase “added to” is deleted.
Page 3 line 95: “was obtained through centrifuged at 10,000 rpm” ” Please, rephrase this part of the sentence (e.g.: …was centrifuged at..).
Response: We have rephrased this sentence. “The graphene slurry was centrifuged at…”
Page 3 line 113: “The coatings investigated in this study consisted in the epoxy based without the addition of filler….” Please, rephrase this part of the sentence.
Response: This sentence has been modified. “The investigated coatings in this study consisted of the epoxy without the addition of filler as well as with the incorporation of PPy nanowires and PPy nanowires/graphene as the functional additives.”
Page 3 line 121: “…and the mixture was painted on the pretreated mild steel” Please, inform the readers about the pretreatment of the mild steel.
Response: The pretreatment process of the mild steel is described in Page 2 line 77 now.
It is interesting that the Authors used NaCl solution in the electrochemical experiments as this chemical causes pitting corrosion and not general one.
Response: The 3.5 wt% NaCl solution is generally used as electrolyte to evaluate the anticorrosion performance of coatings through electrochemical workstations.
How can the Authors prove that during the PPy-graphen preparation the graphene was not oxidized? They mention: “The PPy nanowires/graphene nanocomposites (PPyNG) were synthesized through in situ chemical oxidative polymerization.”
Response: The oxidation of graphene is often processed in either the Brodie, Staudenmaier, or Hummers method. These methods all use high concentration of strong oxidant. However, the concentration of APS was low in the process of in situ polymerization of PPy. This approach is similar to the work of other researchers.1-3
In Figure 3 it would be better to use the same scale (instead of 1μm, 200 nm, 300 nm).
Response: The 200 nm scale has been uniformly used in the Figure 3.
Page 7 line 246: “The EPPyNG2 coating had better barrier property compared to neat epoxy and EPPyN, because the well dispersed PPyNG could fill the structural and pinhole porosity of neat epoxy and then inhibited the water penetration.” Though the Authors described and proved the fact that the 2% of PPyNG2 composite coating showed higher anticorrosion activity than the coating with 3%, but they did not give satisfying explanation for this phenomenon. It is not supported by the structure of the PPyNG1 and PPyNG2, which seemed to be very similar shown in the Figure 3 but the PPyNG3 had a different structure that shows the differences in the efficiency is not due to the structure. On page 9 line 274 the Authors say: “The EPPyNG3 coating also exhibited lower Rpore and higher Cc than EPPyNG2, confirming that excess graphene induced the defect coating and reduced corrosion protection.“ This statement specially needs explanation concerning the sentence given on page 9 line 284: “PPyN reacted with steel to form a dense layer of passive oxide film, resulting in substantially reduced penetration of corrosive medias.” One can think that in the presence of more PPyNG the formation of the passive film is more likely.
Response: The conjugation length of EPPyNG2 was longer than those of EPPyN and EPPyNG1, resulting in electrons easier delocalization and in favor of the formation of passivation layers.4, 5 Hence, the EPPyNG2 also exhibited better anticorrosion performance than those of EPPyN and EPPyNG1. Although EPPyNG3 possessed longer conjugation length, its excess graphene content might increase the defects due to influence of polypyrrole nanowires, and probably generated micro galvanic corrosion that showed the corrosion-promotion activity.6 Therefore, the EPPyNG3 displayed wore anticorrosive property compared with EPPyNG2. These statements are added in Page 8 line 248-256 now.
Sahoo, S.; Dhibar, S.; Hatui, G.; Bhattacharya, P.; Das, C. K. Graphene/polypyrrole nanofiber nanocomposite as electrode material for electrochemical supercapacitor. Polymer 2013, 54 (3), 1033-1042. Xu, C.; Sun, J.; Gao, L. Synthesis of novel hierarchical graphene/polypyrrole nanosheet composites and their superior electrochemical performance. Journal of Materials Chemistry 2011, 21 (30), 11253-11258. Zhang, D.; Zhang, X.; Chen, Y.; Yu, P.; Wang, C.; Ma, Y. Enhanced capacitance and rate capability of graphene/polypyrrole composite as electrode material for supercapacitors. Journal of Power Sources 2011, 196 (14), 5990-5996. Montoya, P.; Jaramillo, F.; Calderón, J.; Córdoba de Torresi, S. I.; Torresi, R. M. Evidence of redox interactions between polypyrrole and Fe3O4 in polypyrrole–Fe3O4 composite films. Electrochimica Acta 2010, 55 (21), 6116-6122. Mohammadkhani, R.; Ramezanzadeh, M.; Saadatmandi, S.; Ramezanzadeh, B. Designing a dual-functional epoxy composite system with self-healing/barrier anti-corrosion performance using graphene oxide nano-scale platforms decorated with zinc doped-conductive polypyrrole nanoparticles with great environmental stability and non-toxicity. Chemical Engineering Journal 2020, 382, 122819.
Reviewer 2 Report
The manuscript discussed the corrosion properties of coatings changing the mixing fillers. Improvement of protection is interesting, but there are several questions. Therefore, I recommend its publication after the revision. Comments are listed below;
The first few sentences in the introduction and conclusion mean nothing to readers. It is too difficult to understand why these sentences are put in the manuscript.
The ratio of 0.5% was determined using EPPyN. Is this ratio best for other conditions? If EPPyNG1, EPPyNG2, or EPPyNG3 is not best with 0.5%, the later discussion becomes meaningless. Therefore, the authors should show that 0.5% is best for all conditions.
Scales and conditions in Fig 3 are not clear.
Although OCP recovered several times in Fig 4, what is the reason for corrosion recovery? Generally, the damage monotonically progresses. Thus, the readers see these results, they might believe that OCP cannot correctly explain the damage progression.
Only a little difference of pyrrole:graphene ratio changes corrosion condition dramatically, which means the mixing of nanocomposites is not always the best way to improve the corrosion but sometimes it becomes the cause of accelerated corrosion. The authors should mention why these differences arise and what is the main factor in preventing corrosion.
Although the authors indicated the difference between PPyNG1,2 and PPyNG3 is granular morphology in PPyNG3, is it the only difference? Also, is there no granular morphology in the entire area for PPyNG1,2? Isn't it just incidentally captured in Fig 3d? Because the observation range of SEM is very narrow, the authors should show evidence that the difference among them is not an accidental result.
The results of EIS (Fig 8) shows the overall trend of corrosion. However, individual trends are different from other results; EPPyNG1 shows a similar trend as EPPyNG2, which indicates that there is a limit to explain the corrosion only from an electrical point of view. Is there any evidence that the corrosion was only due to an electrochemical reaction? If other reactions are also related, the results should be more carefully discussed.
Figure 8 is broken (in the range of 5-10 days), so please fix it.
Although the corrosion protection mechanism is discussed basing that the nanocomposites are evenly distributed in the epoxy, is it true? When the nanocomposites are not distributed, other mechanisms should be dominant. Because there is no discussion about the material properties of EPPyNGs but only the corrosion test, nobody can understand the relation between EPPyNG itself and corrosion property (not between PPyNG and corrosion property). At least, some evidence to show homogeneous distribution of the nanocomposites in the epoxy, e.g. SEM images, should be required.
Author Response
Dear Reviewer 2:
Special thanks to you for your good comment. Your comments are all valuable and very helpful for revising and improving our paper, as well as the important guiding significance to our researchers. We have studied comments carefully and have made correction. The amendments are highlighted in yellow in the revised manuscript. We would like to resubmit it for your kind consideration. Point by point responses to your comments are listed below this letter.
We hope that the revised version of the manuscript is now acceptable for publication. Looking forward to hearing from you.
Thank you and best regards.
Yours sincerely,
Jiang Zhong, Ph. D.
jiangzhong@jxstnu.com.cn
The manuscript discussed the corrosion properties of coatings changing the mixing fillers. Improvement of protection is interesting, but there are several questions. Therefore, I recommend its publication after the revision. Comments are listed below;
The first few sentences in the introduction and conclusion mean nothing to readers. It is too difficult to understand why these sentences are put in the manuscript.
Response: These sentences were in the template of Polymers Journal. Our carelessness results in their appearance in the manuscript. We are very sorry for this. Now we’ve deleted them.
The ratio of 0.5% was determined using EPPyN. Is this ratio best for other conditions? If EPPyNG1, EPPyNG2, or EPPyNG3 is not best with 0.5%, the later discussion becomes meaningless. Therefore, the authors should show that 0.5% is best for all conditions.
Response: This ratio was best for our experimental condition. Three weight concentrations of additives were conducted as shown in Supplementary Materials. The formulations of fillers in EPPyN EPPyNG1, EPPyNG2, EPPyNG3 were all fixed at 0.5 %.
Scales and conditions in Fig 3 are not clear.
Response: Thank you for your careful review. The 200 nm scales are added to the Figure 3. These images in Figure 3 are original, leading to the result that the words of experiment condition are relatively small after picture combination.
Although OCP recovered several times in Fig 4, what is the reason for corrosion recovery? Generally, the damage monotonically progresses. Thus, the readers see these results, they might believe that OCP cannot correctly explain the damage progression.
Response: In Fig. 4, the majority of OCP values were increased with the immersion time. A few slight increases might be due to the accumulation for corrosion products, which was also reported by other researchers.1 We have added a description of this results in the manuscript.
Only a little difference of pyrrole:graphene ratio changes corrosion condition dramatically, which means the mixing of nanocomposites is not always the best way to improve the corrosion but sometimes it becomes the cause of accelerated corrosion. The authors should mention why these differences arise and what is the main factor in preventing corrosion.
Response: The EPPyNG2 coating had better barrier property compared to neat epoxy, because the well dispersed PPyNG could fill the structural and pinhole porosity of neat epoxy and then inhibited the water penetration. The conjugation length of EPPyNG2 was longer than those of EPPyN and EPPyNG1, resulting in electrons easier delocalization and in favor of the formation of passivation layers.2,3 Hence, the EPPyNG2 also exhibited better anticorrosion performance than those of EPPyN and EPPyNG1. Although EPPyNG3 possessed longer conjugation length, its excess graphene content might increase the defects due to influence of polypyrrole nanowires, and probably generated micro galvanic corrosion that showed the corrosion-promotion activity.4 Therefore, the EPPyNG3 displayed wore anticorrosive property compared with EPPyNG2.
Although the authors indicated the difference between PPyNG1,2 and PPyNG3 is granular morphology in PPyNG3, is it the only difference? Also, is there no granular morphology in the entire area for PPyNG1,2? Isn't it just incidentally captured in Fig 3d? Because the observation range of SEM is very narrow, the authors should show evidence that the difference among them is not an accidental result.
Response: Although the observation range of SEM is very narrow, our sampling points for 30 k magnification SEM images were random, and we also randomly took 30 k magnification images in two places. These images are shown in the following figure. It’s a pity that low magnification SEM images were not taken for PPyNG1, PPyNG2 and PPyNG3. However, the random images of two places could also explain our results. In addition, because the other reviewers ask me to put the SEM images in the same scale, we modify the Figure 3.
Figure R1. SEM images of PPyNG1 (a and b), PPyNG2 (c and d), and PPyNG3 (e and f). (Please see the Word Document)
The results of EIS (Fig 8) shows the overall trend of corrosion. However, individual trends are different from other results; EPPyNG1 shows a similar trend as EPPyNG2, which indicates that there is a limit to explain the corrosion only from an electrical point of view. Is there any evidence that the corrosion was only due to an electrochemical reaction? If other reactions are also related, the results should be more carefully discussed.
Response: Individual trends are different from other results probably due to the formation of corrosion products. Our anticorrosive coating is composed of epoxy resin containing polypyrrole/graphene composites. These materials are stable in NaCl solution. Hence, the reaction of our system may be mainly the corrosion of metal and electrochemical reaction between metal and polypyrrole at the interface.
Figure 8 is broken (in the range of 5-10 days), so please fix it.
Response: The test samples with electrochemical measurement were soaked for 3 days, 7days, 15 days, 25 days and 30 days, respectively. Therefore, in Figure 8, a break is seen.
Although the corrosion protection mechanism is discussed basing that the nanocomposites are evenly distributed in the epoxy, is it true? When the nanocomposites are not distributed, other mechanisms should be dominant. Because there is no discussion about the material properties of EPPyNGs but only the corrosion test, nobody can understand the relation between EPPyNG itself and corrosion property (not between PPyNG and corrosion property). At least, some evidence to show homogeneous distribution of the nanocomposites in the epoxy, e.g. SEM images, should be required.
Response: Thanks for your advice. The dispersion of fillers with similar structure in the resin substrate depends mainly on the proportion of fillers to the resin. In our work, the ratio of PPyN or PPyNG to resin were same. The fillers could be uniformly dispersed at this ratio. We have added the optical microscope pictures in the Supplementary Materials Figure S4.
Qiu, S.; Li, W.; Zheng, W.; Zhao, H.; Wang, L. Synergistic Effect of Polypyrrole-Intercalated Graphene for Enhanced Corrosion Protection of Aqueous Coating in 3.5% NaCl Solution. ACS Applied Materials & Interfaces 2017, 9 (39), 34294-34304. Montoya, P.; Jaramillo, F.; Calderón, J.; Córdoba de Torresi, S. I.; Torresi, R. M. Evidence of redox interactions between polypyrrole and Fe3O4 in polypyrrole–Fe3O4 composite films. Electrochimica Acta 2010, 55 (21), 6116-6122. Mohammadkhani, R.; Ramezanzadeh, M.; Saadatmandi, S.; Ramezanzadeh, B. Designing a dual-functional epoxy composite system with self-healing/barrier anti-corrosion performance using graphene oxide nano-scale platforms decorated with zinc doped-conductive polypyrrole nanoparticles with great environmental stability and non-toxicity. Chemical Engineering Journal 2020, 382, 122819. Irfan, M.; Bhat, S. I.; Ahmad, S. Reduced Graphene Oxide Reinforced Waterborne Soy Alkyd Nanocomposites: Formulation, Characterization, and Corrosion Inhibition Analysis. ACS Sustainable Chemistry & Engineering 2018, 6 (11), 14820-14830.

Reviewer 3 Report
The study reported in the paper “Protection of mild steel by waterborne epoxy coatings incorporation of polypyrrole nanowires/graphene nanocomposites” focuses on the production of nanocomposite coatings possessing anticorrosion properties suitable for application to metals, in particular mild steel. The nanocomposites were based on waterborne epoxy resins with the addition of polypyrrole nanowires/graphene (PPyNG) compound produced in different ratios.
The study, although interesting from a scientific point of view, reveals a certain carelessness.
The Authors are requested to improve the manuscript, as following detailed.
- Introduction section: the Authors are requested to delete the lines 28-35.
- Materials section: the Authors are requested to correct all the chemical formula (example: KMnO4 instead of KMnO4, H2SO4 instead of H2SO4, etc.). It is necessary to add also the purchase information for all chemicals.
- It is strongly suggested to report in a Table the composition of all the systems produced for clarity sake.
- Characterization section: the Authors are requested to specify on what systems (PPyNG or EPPyNG) the characterization tests were carried out and to supply details on the specimens analyzed in each test. In addition, it is necessary to state, from the Introduction section, if with “nanocomposites” they refer to PPyNG or to the epoxy coatings containing PPyNG.
- FTIR results: the Authors are requested to correct everywhere “cm−1” in “cm−1”.
- In SEM images, the scale bar is not easily readable. It is suggested to report all images in the same scale in order to allow comparisons between the different pictures.
- Conclusions: the Authors are requested to delete the lines 91-93.
- the Authors are requested to carefully revise the manuscript in order to correct all the typos (as a few examples: at line 103 “mmol” should be “mol”, at line 101 “EPPyNG nanocomposites coatings” should be “EPPyNG nanocomposite coatings”)
Author Response
Dear Reviewer 3:
Special thanks to you for your good comment. Your comments are all valuable and very helpful for revising and improving our paper, as well as the important guiding significance to our researchers. We have studied comments carefully and have made correction. The amendments are highlighted in yellow in the revised manuscript. We would like to resubmit it for your kind consideration. Point by point responses to your comments are listed below this letter.
We hope that the revised version of the manuscript is now acceptable for publication. Looking forward to hearing from you.
Thank you and best regards.
Yours sincerely,
Jiang Zhong, Ph. D.
jiangzhong@jxstnu.com.cn
The study reported in the paper “Protection of mild steel by waterborne epoxy coatings incorporation of polypyrrole nanowires/graphene nanocomposites” focuses on the production of nanocomposite coatings possessing anticorrosion properties suitable for application to metals, in particular mild steel. The nanocomposites were based on waterborne epoxy resins with the addition of polypyrrole nanowires/graphene (PPyNG) compound produced in different ratios.
The study, although interesting from a scientific point of view, reveals a certain carelessness.
The Authors are requested to improve the manuscript, as following detailed.
- Introduction section: the Authors are requested to delete the lines 28-35.
Response: We apologize for this confusion sentences caused by our carelessness. We’ve deleted these redundant sentences.
- Materials section: the Authors are requested to correct all the chemical formula (example: KMnO4 instead of KMnO4, H2SO4 instead of H2SO4, etc.). It is necessary to add also the purchase information for all chemicals.
Response: Thanks for your careful review. We have corrected all the chemical formula and added missing chemicals purchase information.
- It is strongly suggested to report in a Table the composition of all the systems produced for clarity sake.
Response: Thanks for your advice. Results of |Z|0.01Hz with immersion time are summarized in Table 2.
- Characterization section: the Authors are requested to specify on what systems (PPyNG or EPPyNG) the characterization tests were carried out and to supply details on the specimens analyzed in each test. In addition, it is necessary to state, from the Introduction section, if with “nanocomposites” they refer to PPyNG or to the epoxy coatings containing PPyNG.
Response: We have added the sample information to characterize. In addition, the reference to “nanocomposite” and “nanocomposite coatings” has been explained in the introduction section line 65.
- FTIR results: the Authors are requested to correct everywhere “cm−1” in “cm−1”.
Response: We have checked the full text and corrected all the characters that should be superscript or subscript.
- In SEM images, the scale bar is not easily readable. It is suggested to report all images in the same scale in order to allow comparisons between the different pictures.
Response: The same scale have been used in the SEM images.
- Conclusions: the Authors are requested to delete the lines 91-93.
Response: We are very sorry for the appearance of these unrelated sentences. We have deleted them.
- the Authors are requested to carefully revise the manuscript in order to correct all the typos (as a few examples: at line 103 “mmol” should be “mol”, at line 101 “EPPyNG nanocomposites coatings” should be “EPPyNG nanocomposite coatings”)
Response: Thanks for your careful review. The “mmol” represents that the amount of APS is millimole. In addition, we have corrected the typos “EPPyNG nanocomposites coatings” with “EPPyNG nanocomposite coatings”.

Round 2
Reviewer 2 Report
The manuscript is well improved and I have no more comment.
Reviewer 3 Report
The authors took into account the suggestions of this referee, modifying the text accordingly and providing appropriate answer in their letter. In my opinion the paper is now suitable for publication.